# Cost-effectiveness of varicella and herpes zoster vaccination in Sweden: An economic evaluation using a dynamic transmission model

Ellen Wolff[1,2]*, Katarina Widgren[1,3], Gianpaolo Scalia Tomba[4], Adam Roth[5], Tiia Lep[1], Sören Andersson[1]

1 Department of Public Health Analysis and Data Management, Public Health Agency of Sweden, Solna, Sweden, 2 School of Public Health and Community Medicine, Institute of Medicine, University of Gothenburg, Göteborg, Sweden, 3 Department of Medicine, Huddinge C2:94, Karolinska University Hospital, Stockholm, Sweden, 4 Department of Mathematics, University of Rome Tor Vergata, Rome, Italy, 5 Institution for Translational Medicine, Lund University, Malmö, Sweden

* ellen.wolff@folkhalsomyndigheten.se

**Data Availability Statement:** All relevant data are within the paper.

**Funding:** The authors received no specific funding for this work.

## Abstract

### Objectives

Comprehensive cost-effectiveness analyses of introducing varicella and/or herpes zoster vaccination in the Swedish national vaccination programme.

### Design

Cost-effectiveness analyses based on epidemiological results from a specifically developed transmission model.

### Setting

National vaccination programme in Sweden, over an 85- or 20-year time horizon depending on the vaccination strategy.

### Participants

Hypothetical cohorts of people aged 12 months and 65-years at baseline.

### Interventions

Four alternative vaccination strategies; 1, not to vaccinate; 2, varicella vaccination with one dose of the live attenuated vaccine at age 12 months and a second dose at age 18 months; 3, herpes zoster vaccination with one dose of the live attenuated vaccine at 65 years of age; and 4, both vaccine against varicella and herpes zoster with the before-mentioned strategies.

**Competing interests:** The authors have declared that no competing interests exist.

## Main outcome measures

Accumulated cost and quality-adjusted life years (QALY) for each strategy, and incremental cost-effectiveness ratios (ICER).

## Results

It would be cost-effective to vaccinate against varicella (dominant), but not to vaccinate against herpes zoster (ICER of EUR 200,000), assuming a cost-effectiveness threshold of EUR 50,000 per QALY. The incremental analysis between varicella vaccination only and the combined programme results in a cost per gained QALY of almost EUR 1.6 million.

## Conclusions

The results from this study are central components for policy-relevant decision-making, and suggest that it was cost-effective to introduce varicella vaccination in Sweden, whereas herpes zoster vaccination with the live attenuated vaccine for the elderly was not cost-effective–the health effects of the latter vaccination cannot be considered reasonable in relation to its costs. Future observational and surveillance studies are needed to make reasonable predictions on how boosting affects the herpes zoster incidence in the population, and thus the cost-effectiveness of a vaccination programme against varicella. Also, the link between herpes zoster and sequelae need to be studied in more detail to include it suitably in health economic evaluations.

## Introduction

The varicella-zoster virus (VZV) causes both varicella (chickenpox) and herpes zoster (shingles). Varicella is the clinical presentation of primary infection with the varicella-zoster virus. Due to the high contagiousness of varicella, nearly everyone will contract the disease early in life. A study from 1997 showed that 98% of Swedish 12-year olds had VZV IgG antibodies, i.e. had had varicella at some time point before that age [1]. Varicella is generally a mild disease in children that lasts about a week [2], but complications can occur. The risk of severe disease is higher in adults and immunocompromised persons [2]. Effective vaccines against varicella have been in use since the mid-1990s [3] and routine childhood vaccination programmes are in place in several countries worldwide, for instance, the United States since 1995 and later, Canada, Australia, Germany and Finland [4, 5]. Significant declines in varicella incidence after the introduction of the vaccine have been observed [5–8].

Following the primary varicella infection, the virus remains latent in dorsal root and cranial nerve ganglia. The virus can reactivate, often decades later, which leads to herpes zoster. In the current Swedish setting where nearly everyone has contracted varicella early in life, almost all adults are at risk of developing herpes zoster. The lifetime risk is estimated to 25–30%, and in the age group 85 and older, the risk is as high as 50% [9]. The manifestation of herpes zoster is usually a unilateral vesicular rash in the skin area supplied by the affected nerve accompanied by itching, pain, and/or numbness. The pain may be intense and is described as burning or electric-like pain that usually resolves within 2 to 4 weeks. Complications are relatively common, such as post-herpetic neuralgia (PHN), or non-PHN complications as meningitis, encephalitis, and other complication from the nervous system. There are currently two

vaccines against herpes zoster; one live attenuated vaccine and one adjuvanted subunit recombinant vaccine. However, as of March 2020, the subunit vaccine is not yet available in Sweden apart from a few doses offered on the private market.

In Sweden, neither varicella nor herpes zoster are part of a national vaccination programme. Vaccines against varicella and herpes zoster are available in Sweden at an out-of-pocket expense for the individual. This study aimed to perform comprehensive cost-effectiveness analyses of an introduction of varicella and/or herpes zoster vaccination in the Swedish national vaccination programme. The cost-effectiveness analyses were based on the epidemiological results from a specifically developed transmission model. In this model, a more moderate approach to include exogenous boosting was chosen than in previous studies.

## Methods

We linked an age-structured dynamic Markov transmission model of the varicella-zoster virus (VZV) with costs and health effects of the diseases, to investigate the potential cost-effectiveness of different vaccination programmes against varicella and/or herpes zoster in Sweden. The model used age-structured social contact patterns to simulate at-risk contacts and incorporated different ways of contracting the disease, such as natural infection and reactivation, breakthrough infection and vaccine-derived disease. A dynamic model was developed to account both for the direct effects of vaccination but also the indirect effects of vaccination such as herd-immunity.

Ethical approval was not required, and the manuscript is an honest, accurate, and transparent account of the study being reported.

## Model

### The epidemiological model

We developed an epidemiological transmission model to explore the transmission of VZV in Sweden by mathematical modelling, and the impact of vaccination programmes targeting varicella and/or herpes zoster was assessed. The transmission model was developed in the programming language C. The outcome data from the transmission model were extracted to Excel where the health economic model was developed.

The transmission model was a deterministic, age-structured, dynamic Markov model. The dynamics of the model were represented as differential equations with a one-day time cycle. Individuals could transfer from one health state to another, stay in the same health state or die in each time-cycle. The population data were obtained from Statistics Sweden as of 2012. The contact patterns between age groups were based on a synthetic matrix by Fumanelli et al. [10] that describes the intensity of total contacts between age groups and helps us model transmission of disease. Boosting was modelled as a form of complete immunity to herpes zoster that wanes with time (a modified temporary immunity model) but without cumulative increasing protection [11].

A simplified flowchart is presented in Fig 1.

Individuals were born into the model, and were protected against infection by maternal antibodies for an average of three months at birth and then moved to the health state "susceptible to varicella". If infected, the average incubation period for varicella was set to 14 days and the average duration of contagiousness to 7 days [12]. After recovering from varicella, individuals transferred to the health state "susceptible to herpes zoster" and could, at an age-dependent rate, develop herpes zoster. If boosted, individuals became temporarily protected from herpes zoster, after which they returned to the susceptible state. When vaccination was introduced into the model, a proportion, corresponding to the varicella vaccination coverage,

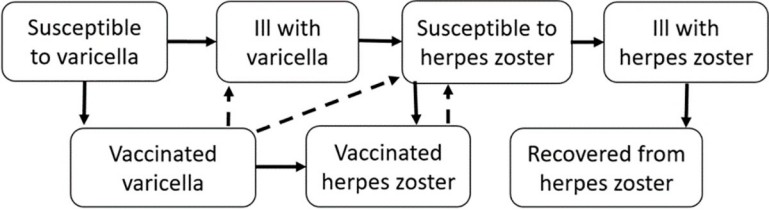

**Fig 1. Simplified flowchart of the epidemiological model.**

moved from the "susceptible to varicella" state to the health state "vaccinated against varicella" when the vaccination was given. If vaccination failed or waned, a vaccinated person could become infected and develop so-called breakthrough varicella if exposed. Varicella vaccinated individuals were at risk to develop vaccine-strain herpes zoster, and those who developed breakthrough varicella were also at risk of reactivation of the wild-type zoster virus.

The health states were mutually exclusive. Age-specific incidence was applied to the susceptible population, and population mortality acted equally in all states. Costs and health effects were applied to each health state and aggregated annually.

At 65 years of age, in the case of vaccination against herpes zoster, a proportion corresponding to the herpes zoster vaccination coverage moved to the health state "vaccinated against herpes zoster". The protection of the vaccine waned over time and vaccinated individuals could therefore move from the vaccinated state to the state "susceptible to herpes zoster".

Recent review papers have concluded that although a boosting effect of immunity when exposed to VZV probably exists, the magnitude on a population level is uncertain [13–15]. In addition, the experience from countries with long-lasting routine varicella vaccination, such as recent surveillance data from e.g. USA [16, 17] speaks against a strong surge in herpes zoster incidence after implementing a varicella vaccination programme for children. Given this, we have chosen a more moderate approach to include exogenous boosting, making boosting comparable to a live herpes zoster vaccination of limited duration, similarly to Melegaro et al [18].

**Incidence of varicella and herpes zoster.** *Varicella.* The age-dependent varicella incidence was obtained from combining Swedish seroprevalence data with Finnish seroprevalence data, from before varicella was in the national vaccination programme in Finland [19]. The resulting annual number of varicella cases was approximately 118,000 without a vaccination programme, compared to the assumed stable yearly birth cohort of 120,500. For the baseline varicella hospitalization incidence in the health economic model, we used national register-based study by Widgren et al [20].

*Herpes zoster.* The baseline herpes zoster incidence, after the first contact with the virus, was assumed to be age-dependent in the transmission model. Regional data (Region Västra Götaland) on consultations in primary, specialist and in-patient health care that covered the years 2008 to 2010 were used to estimate baseline incidence [21]. The annual number of herpes zoster cases for the whole country was estimated to approximately 30,000 without a vaccination programme. This was equivalent of a 25% lifetime risk of developing herpes zoster. The risk of reactivating vaccine strain virus after varicella vaccination was assumed to be 10% of the reactivation rate of wild-type virus [22–26]. We assumed that 12.5% of individuals with herpes zoster develop PHN [27] and that 10% develop non-PHN complications.

**Vaccine effectiveness.** The vaccine effectiveness of the first dose of the varicella vaccine was set to 81%, and 92% for the second dose, in the epidemiological model. After the first dose, we assumed a waning rate of 2% per year. No waning was assumed after the second dose [28–31].

In the case of herpes zoster vaccination at 65 years of age, the effectiveness of live herpes zoster vaccine was assumed to be 64%, with an average duration of four years and significant waning after eight years, when only 14% still have immunity [32]. The vaccine effectiveness against PHN was somewhat higher, 73% [33], no additional waning was applied on the effectiveness against PHN [34–36]. The vaccine effectiveness of the recombinant herpes zoster vaccine was assumed to be 97% among 65-year-olds with an average duration of 8 years [32, 37, 38].

**Vaccination strategies.** We have investigated and simulated four alternative vaccination strategies in the Swedish context: firstly, not to vaccinate; secondly, varicella vaccination only with one dose of the live attenuated vaccine at age 12 months and a second dose at age 18 months; thirdly, herpes zoster vaccination only with one dose of the live attenuated vaccine at 65 years of age; and finally both vaccination against varicella and herpes zoster with the above-mentioned strategies.

We assumed a vaccination coverage of varicella vaccination of 95%. This is in line with the vaccination coverage in the national vaccination programme for children in Sweden [39]. The vaccination coverage for herpes zoster vaccination was assumed to be 50%, based on the coverage of seasonal influenza vaccination in the equivalent age group.

## Health economic evaluation

The cost-effectiveness of the vaccination programmes was assessed by accumulating costs and health effects over time to create an incremental cost-effectiveness ratio (ICER) of the vaccination programme evaluated compared with no vaccination. An incremental cost-effectiveness ratio describes the cost for society to gain an additional quality-adjusted life-year (QALY) and is calculated as the difference in costs divided with the difference in health effects. For a meaningful comparison, the additional costs and health effects that one programme imposes over another was also examined, i.e. an incremental analysis of the independent strategies, in addition to the comparison with no vaccination. This was conducted between varicella vaccination only and the combined programme with both varicella and herpes zoster vaccination.

To include all relevant effects of the vaccination, the model was simulated over a lifetime horizon; varicella vaccination was evaluated in an 85 years perspective and herpes zoster vaccination was evaluated in the 20 years perspective. The different time horizons were due to the different age at vaccination. When evaluating the vaccination programme including both varicella and herpes zoster vaccination the time horizon was set to 85 years, to take into account the effect of varicella vaccination on herpes zoster incidence.

Health effects and costs were discounted by 3% annually, according to the guidelines for health economic evaluations conducted in a Swedish context [40]. Results were also presented without discounting in sensitivity analyses as well as with differential discounting, which is recommended in a European standard for health economic analyses of vaccination programs [41].

**Resource use.** *Varicella*. The varicella vaccine was given in two doses, and we assumed, together with clinical expertise, that the resource use of administering the vaccine was 15 minutes per dose. We assumed that two nurses were needed to work together with each patient to give the first dose since multiple vaccines were given at the same time [39].

The resource use that was applied in the model for individuals ill with varicella is shown in Table 1 and is presented as the share of patients needing primary care, specialist care and hospitalization. The data were primarily obtained from a study by Widgren et al [20]. When data were missing, clinical expertise was consulted. We assumed that breakthrough infection gave rise to a milder varicella disease than natural infection and that individuals with breakthrough varicella did not need hospitalisation [29].

**Table 1. Resource use.** Resource use for or varicella and herpes zoster applied in the model, the share of cases in each health care setting, by age group.

| | Age group | Share primary care (%) | Share specialist care (%) | Share hospitalized (%) | Average no. days hospitalized | Average no. days with production loss | Source |
|---|---|---|---|---|---|---|---|
| Varicella | 0 | 3 | 1,9 | 0,2 | 3,7 | 2,4 | [20] |
| | 1 | 13 | 2,5 | 0,4 | 3 | 3,57 | [20] |
| | 2 | 17 | 2,3 | 0,4 | 3,7 | 3,61 | [20] |
| | 3 | 8 | 1 | 0,2 | 4,3 | 3,52 | [20] |
| | 4 | 6 | 0,7 | 0,1 | 3,3 | 3,5 | [20] |
| | 5 | 7 | 0,8 | 0,2 | 3,9 | 3,46 | [20] |
| | 6 | 6 | 0,7 | 0,1 | 4,2 | 3,46 | [20] |
| | 7 | 7 | 0,9 | 0,2 | 9,3 | 3,48 | [20] |
| | 8 | 6 | 0,6 | 0,1 | 10,2 | 3,44 | [20] |
| | 9 | 5 | 0,8 | 0,1 | 20,6 | 3,58 | [20] |
| | 10 | 7 | 1,1 | 1 | 3,9 | 3,45 | [20] |
| | 11 | 7 | 1,1 | 1 | 3,9 | 3,43 | [20] |
| | 12 | 7 | 1,2 | 1,1 | 3,9 | 3,76 | [20] |
| | 13 | 7 | 1,2 | 1,2 | 3,9 | 4,65 | [20] |
| | 14 | 8 | 1,4 | 1,3 | 3,9 | 3 | [20] |
| | 15–24 | 19 | 4,7 | 1 | 8,2 | 2,5 | [20] and expert opinion |
| | 25–44 | 23 | 6,7 | 1 | 3,2 | 5 | [20] and expert opinion |
| | 45–64 | 27 | 5,1 | 3 | 11 | 5 | [20] and expert opinion |
| | 65+ | 27 | 10,6 | 6 | 11,4 | - | [20] and expert opinion |
| Herpes zoster | 0–64 | 97 | - | 3 | 7,5 | 10 | [42] and expert opinion |
| | 65+ | 97 | - | 3 | 7,5 | - | [42] |

*Herpes zoster*. The live herpes zoster vaccine was given in one dose, and we assumed that 80% of those being vaccinated needed an extra visit for the administration of the vaccine, instead of receiving the dose at a regular visit to the doctor's office. The recombinant vaccine was available on the Swedish private market as of April of 2020. However, there will only be a smaller number of doses available and was currently considered not relevant for a national vaccination programme.

The resource use that was applied for individuals ill with herpes zoster is partially shown in Table 1. The specifics is further explained in this section. About 94% of all patients with wild-type herpes zoster were assumed to be treated with antivirals in all age groups [42] Out of those ill with herpes zoster, 97% were estimated to have one visit to primary care, and 3% in need of hospitalization. If hospitalized, the patient stayed for an average of 7.5 days at the hospital [42]. We assumed that reactivation of varicella vaccine-strain virus gave rise to a milder herpes zoster disease than reactivation of a wild-type virus and that individuals with vaccine-strain herpes zoster did not need hospitalisation.

In addition, individuals with herpes zoster (all causes) could develop either post-herpetic neuralgia (PHN) or non-PHN complications. In the economic model, we assumed that 12.5% of all individuals with herpes zoster developed PHN [27] and that they were treated with Gabapentin (300 mg, 3 times daily) for nine months [43], and had three additional primary care visits. .10% of all individuals with herpes zoster were assumed to develop non-PHN complications. This was assumed in collaboration with clinical experts, and non-PHN

**Table 2. Cost per item used in the health economic model.**

| Item* | Comment | Price (EUR) | Source |
|---|---|---|---|
| Vaccine, varicella | 2 doses | 45 | [45] |
| Vaccine live, herpes zoster | 1 dose | 125 | [46] |
| Administration of vaccine | VA000, 2014 | 17 | [44] |
| Antivirals | One package, 42 pills a 500 g | 12 | [47] |
| Gabapentin | Three packages, 100 pills a 300 mg | 34 | [48] |
| Weighted average cost for non-PHN complications | ICD-10 B02.0, B02.1, B02.2, B02.3 | 4,844 | [49] |
| Cost per stroke case | Assumed to be hospitalized for an average of 5.6 days | 4 250 | [49] |
| Child health care nurse (administration cost for administrating vaccine for children) | Average hourly wage | 24 | [50] |
| Doctors visit (GP) | | 205 | [44] |
| Doctors visit for children | | 355 | [44] |
| Admission cost, ward | | 290 | [44] |
| Admission cost for doctor | | 178 | [44] |
| Cost of doctor, per hospital day | | 134 | [44] |
| Cost per hospital day | | 571 | [44] |
| Admission cost, ward for children | | 75 | [44] |
| Admission cost for a doctor, for children | | 94 | [44] |
| Cost of doctor, per hospital day, for children | | 231 | [44] |
| Cost per hospital day, for children | | 831 | [44] |
| Production loss per day | | 212 | [51] |

*Item refers to the resource use. For example, "Admission cost for doctor" refers to the cost each patient give rise to when a doctor examines them at admission to a hospital ward

complications were defined as ICD codes: B02.0, B02.1, B02.2, and B03.3. We also accounted for stroke as a complication of herpes zoster and used data from Sundström et al [21] to estimate resource use of stroke caused by herpes zoster. In the economic model, 0.4% of individuals with herpes zoster were assumed to suffer a stroke within one year of herpes zoster diagnosis.

**Costs.**  We used the list prices for estimating the cost of both varicella and herpes zoster vaccination since the procurement prices are not official in Sweden and could change over time at each new procurement process.

The costs of resource use in the health care sector was mostly obtained from a price list from the southern hospital region in Sweden, see Table 2 [44].

The cost of resource use for non-PHN complications of herpes zoster was obtained from the cost per patient database (ICD-10 codes: B02.0, B02.1, B02.2, and B03.3) [49]. We used a weighted average of both primary care and hospitalization.

The health economic analysis had a societal perspective in the base-case analysis, implying that indirect costs were included in the analysis in the form of production losses in case of illness. Production loss was included for individuals 0–65 years old in the model. Individuals older than 65 years old were assumed to be retired, and for children production loss was calculated as the number of days a parent needed to be home with his or her sick child. The cost of productivity losses was calculated based on the average monthly salary of EUR 3,383 [52] and the statutory employers' fee of 31.42% [53]. This inferred a productivity loss of EUR 4,446 per month or EUR 212 per working day. The length of the productivity loss in the model was dependent on disease and age group (see Table 1).

All costs were measured in Swedish krona (SEK), converted to Euro (exchange rate 100 SEK = 10.82 euro [2020-04-28, Swedish Central Bank]), and measured at 2018 price level.

**Table 3. QALY loss, dependent on disease and age group [54].**

| | Varicella | | Herpes zoster |
|---|---|---|---|
| | **Natural** | **Breakthrough** | |
| 0–14 | 0,0027 | 0,0014 | 0,022 |
| 15–34 | 0,0038 | 0,0019 | 0,022 |
| 35–44 | 0,0038 | 0,0019 | 0,027 |
| 40–54 | 0,0038 | 0,0019 | 0,032 |
| 55–64 | 0,0038 | 0,0019 | 0,0495 |
| 60–74 | 0,0038 | 0,0019 | 0,067 |
| 75–84 | 0,0038 | 0,0019 | 0,134 |
| 85+ | 0,0038 | 0,0019 | 0,201 |

**Health effects.** Health-related effects of varicella-related disease were calculated as QALY-losses, i.e. multiplying the number of individuals in a health state with the associated QALY-loss. The QALY-loss associated with the different health states is presented in Table 3 and were obtained from a study by van Hoek et al [54].

## Sensitivity analyses

Several sensitivity analyses were performed, varying the values of the ingoing variables in the model, to explore how the results were affected by this. Both epidemiological and economic variables were varied to explore how an alteration in the variables would affect the results, both upwards and downwards. To take into account the potential rebates that could be negotiated between regions and vaccine producers, so-called procurement prices, we conducted sensitivity analyses where we studied the effect of the vaccine price on the cost per gained QALY (ICER), given the assumptions made in the model. This analysis was only conducted if the cost per gained QALY was larger than zero.

We varied on the following variables in the sensitivity analyses for the varicella-only vaccination programmes (for ranges see Table 5); the discount rate for costs and health effects (QALY), vaccination coverage, time horizon, vaccine effectiveness, waning rate of the vaccine, reactivation rate, exogenous boosting, administration cost of the vaccine, and an alternative vaccination programme (first dose at 18 months, second dose at 6–8 years). For the herpes zoster vaccination programme, we focused on the discount rate, vaccination coverage, time horizon, duration of PHN, exogenous boosting, and an alternative vaccination programme (vaccinate at age 75). We performed a sensitivity analysis investigating the cost-effectiveness 30–50 years after the implementation of the vaccine, i.e. in a steady state. In addition, we performed a sensitivity analysis where we assumed that the recombinant zoster vaccine was given with two doses, instead of the live vaccine. Since there are only a few doses available on the Swedish market, and the price of those doses are similar to that in the USA, we assumed a price per dose similar to that in the USA, i.e. 144 USD, and the vaccine was given in two doses (exchange rate 100 SEK = 10.82 euro [Swedish Central Bank]).

All other variables were as described above.

## Results

With varicella vaccination, the annual number of cases fluctuated the first 12 years after vaccination was implemented and then stabilized at a very low level of less than 300 cases per year. Fig 2 presents the annual number of cases of varicella, with and without varicella vaccination.

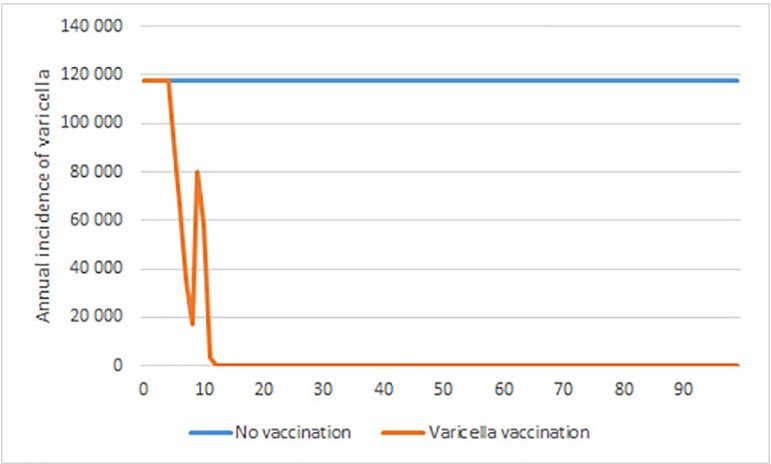

**Fig 2. Annual number of varicella cases with varicella vaccination.** All age groups, with and without varicella vaccination, over a 100-year time horizon.

The number of varicella cases was only marginally affected by herpes zoster vaccination and is not illustrated in the figure.

Fig 3 presents the annual number of herpes zoster cases with and without herpes zoster vaccination. Given the assumptions presented above, herpes zoster vaccination only has a small impact on the annual number of cases in all age groups, and the incidence stabilizes about 10 years after implementation. This was due both to the short duration of the vaccine, the vaccination coverage as well as that vaccination does not affect the zoster incidence in the ages younger than 65-year-old.

Varicella vaccination, as visible in Fig 4, have a great impact on the annual number of herpes zoster cases. The herpes zoster incidence drops from about 30,000 a year to about 4,000 a year over a 100-year time horizon with varicella vaccination. Approximately 20 years into the model, varicella vaccination have a greater impact on the annual number of herpes zoster cases than herpes zoster vaccination does.

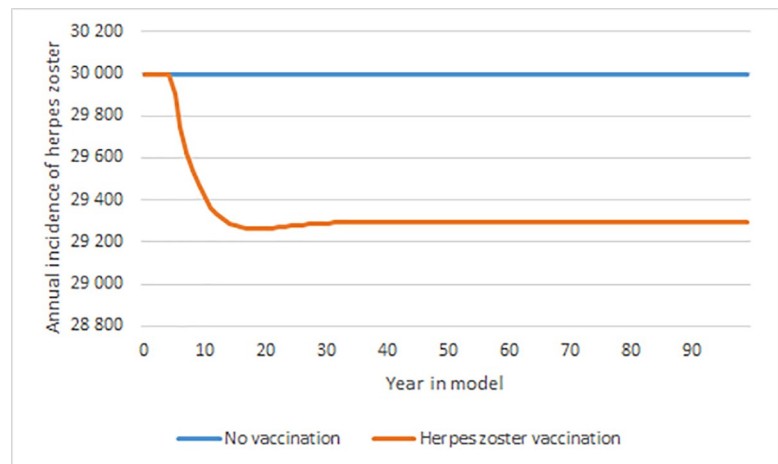

**Fig 3. Annual number of herpes zoster cases with herpes zoster vaccination.** All age groups, with and without herpes zoster vaccination, over a 100-year time horizon. Note the different values on the y-axis compared to Figs 2 and 4 herpes z.

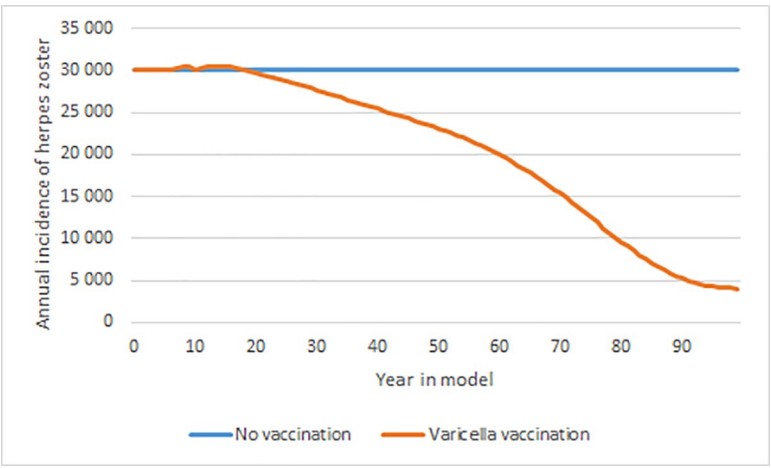

**Fig 4. Annual number of herpes zoster cases with varicella vaccination.** All age groups, with and without varicella vaccination, over a 100-year time horizon.

## Cost-effectiveness

Table 4 shows the total costs of resource use, production losses, and the total QALY losses for the three different vaccination programs as well as in the scenario without vaccination, over the different time horizons. The table also presents the cost per gained QALY for varicella vaccination and herpes zoster vaccination in comparison to no vaccination, as well as the incremental analysis between varicella vaccination only and the combined programme.

Including varicella vaccination in a national vaccination programme would be a dominant strategy compared to no vaccination, i.e. have a better effect at a lower cost. This was true also when the effect on herpes zoster was excluded. From a health care perspective, i.e. without the inclusion of production losses, the routine varicella vaccination programme would still be a dominant strategy. The incremental analysis between varicella vaccination only and the competing strategy of a combined programme with varicella and herpes zoster vaccination results in a cost per gained QALY of almost EUR 1.6 million. The cost per gained QALY for only herpes zoster vaccination was EUR 270,000, and the results remained nearly unchanged with the inclusion of production losses since the majority of the cases were older than 65 years.

In Fig 5, the incremental costs and QALY are presented on the cost-effectiveness plane. Point A indicates the increased costs and QALY with herpes zoster vaccination compared to no vaccination, and the slope of the curve is the cost per gained QALY. Point B is the corresponding figures for varicella vaccination only, and the slope between point B and point C is the cost per gained QALY for the combined programme in comparison with varicella vaccination only.

## Sensitivity analyses

To investigate the robustness of the results from the cost-effectiveness analysis, we conducted several sensitivity analyses. The results are presented in Table 5, with the difference in costs and QALY loss between the investigated scenario and base-case and its related ICER. As visible from the table, the conclusion of the results, i.e. cost-effective/not cost-effective, for varicella vaccination, and herpes zoster vaccination were not affected by varying the before-mentioned parameters.

**Table 4. Results from the cost-effectiveness analyses.** The total costs for health care use, production losses, intervention cost, and the QALY loss for each scenario and its related cost per gained QALY (ICER) in comparison to no vaccination and the incremental analysis between varicella only and the combined programme.

| 85-year time-horizon | Base case (no vaccination) | With varicella vaccination only | With both varicella and herpes zoster vaccination |
|---|---|---|---|
| Intervention cost | - € | 386 361 922 € | 1 261 586 829 € |
| Health care cost | 1 380 321 326 € | 929 065 426 € | 955 856 807 € |
| Varicella | 181 875 147 € | 42 318 704 € | 42 199 135 € |
| Herpes zoster | 1 198 446 179 € | 886 746 722 € | 913 657 672 € |
| Productivity loss | 3 573 888 580 € | 1 337 681 699 € | 1 337 615 736 € |
| Varicella | 2 661 734 488 € | 606 720 558 € | 606 636 554 € |
| Herpes zoster | 912 154 092 € | 730 961 141 € | 730 979 182 € |
| Total cost | 4 954 209 906 € | 2 653 109 047 € | 3 555 059 372 € |
| QALY loss | 79 706 | 69 528 | 68 951 |
| ICER, compared to no vaccination | | Dominant | |
| ICER, both varicella and herpes zoster compared with only varicella vaccination | | | 1 564 923 € |
| 20-year time horizon | Base case (no vaccination) | With herpes zoster vaccination only | |
| Intervention cost | € | 115 642 549 € | |
| Health care cost | 534 733 684 € | 525 620 487 € | |
| Productivity loss | 1 735 828 910 € | 1 735 814 538 € | |
| Total cost | 2 270 562 594 € | 2 377 077 574 € | |
| QALY loss | 38 713 | 38 315 | |
| ICER (compared to base case) | | 267 596 € | |

In Fig 6, the effect on the cost per gained QALY is illustrated at different rebates for the vaccine. The cost per gained QALY decreases by about EUR 26,000 for each 10% increase in the rebate on the vaccine price. Even if the vaccine was free of charge, there remains a small cost per gained QALY due to the cost of administering the vaccine, i.e. vaccination with the live attenuated herpes zoster vaccine would not be cost-saving even if the price of the vaccine would be zero.

We also performed a sensitivity analysis with the recombinant vaccine against herpes zoster. The results from the transmission model indicate a better effect of vaccinating with the recombinant vaccine, than with the live attenuated vaccine. The cost per gained QALY would, assuming a price based on the USA market, be about EUR 170,000, given the assumptions listed in previous sections. An ICER of 170,000 cannot be considered cost-effective.

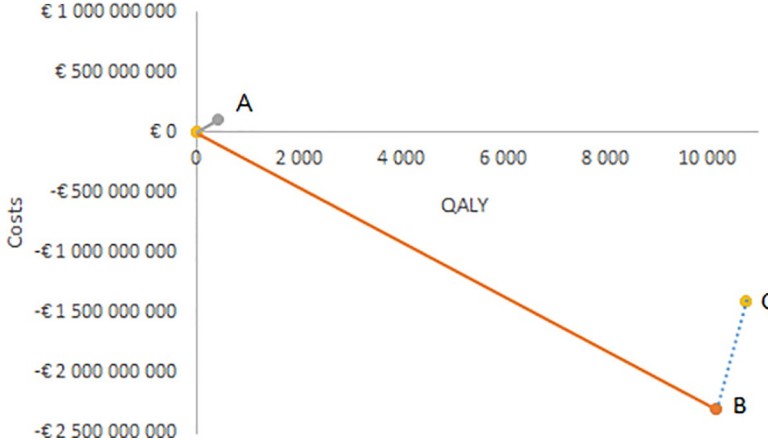

**Fig 5. Incremental cost-effectiveness ratios illustrated on the cost-effectiveness plane.**

**Table 5. Results from the sensitivity analyses for the varicella and herpes zoster vaccination strategies compared to no vaccination, societal perspective.**

| | Difference in costs compared to no vaccination | Difference in QALY compared to no vaccination | ICER |
|---|---|---|---|
| Varicella only | | | |
| Base case | - 2 301 100 859 € | 10 178 | Dominant |
| 0% discounting (health effects and costs) | - 8 170 632 293 € | 44 874 | Dominant |
| 0% discounting (health effects) | - 2 301 100 859 € | 44 874 | Dominant |
| 5% discounting (costs) | - 1 240 709 782 € | 10 178 | Dominant |
| Vaccination coverage 85% | - 2 327 959 003 € | 10 151 | Dominant |
| Time horizon 50 years | - 1 687 437 354 € | 6 237 | Dominant |
| Time horizon 20 years | - 643 900 799 € | 2 304 | Dominant |
| Time horizon 10 years | - 128 874 302 € | 594 | Dominant |
| Vaccine effectiveness 90% both doses | - 2 301 410 283 € | 10 181 | Dominant |
| Waning rate of vaccine 0.5% per year | - 2 300 749 829 € | 10 174 | Dominant |
| Reactivation rate 20% | - 2 295 982 165 € | 9 658 | Dominant |
| No boosting | - 2 367 609 414 € | 14 254 | Dominant |
| No cost of administrating second dose | - 2 322 730 423 € | 10 178 | Dominant |
| 1st dose at 18 months, second at 6–8 years | - 2 300 577 856 € | 10 186 | Dominant |
| Herpes zoster | | | |
| Base case | 106 514 980 € | 398 | 267 596 € |
| 0% discounting (health effects and costs) | 138 035 066 € | 586 | 235 430 € |
| 0% discounting (health effects) | 106 514 980 € | 586 | 181 670 € |
| 5% discounting (costs) | 91 372 771 € | 398 | 229 554 € |
| Vaccination coverage 40% | 107 688 440 € | 318 | 338 261 € |
| Vaccination coverage 60% | 105 339 077 € | 478 | 220 504 € |
| PHN duration 5 years | 103 458 698 € | 398 | 259 917 € |
| Time horizon 50 years | 180 162 628 € | 923 | 195 185 € |
| Time horizon year 30–50 | 41 614 196 € | 287 | 144 895 € |
| No boosting | 125 081 658 € | 1 436 | 87 113 € |
| Vaccine at age 75 years old | 92 184 194 € | 511 | 180 530 € |

## Discussion

We found that it would be cost-effective to vaccinate against varicella, but not to vaccinate against herpes zoster, assuming a cost-effectiveness threshold of 50,000 Euros per QALY from

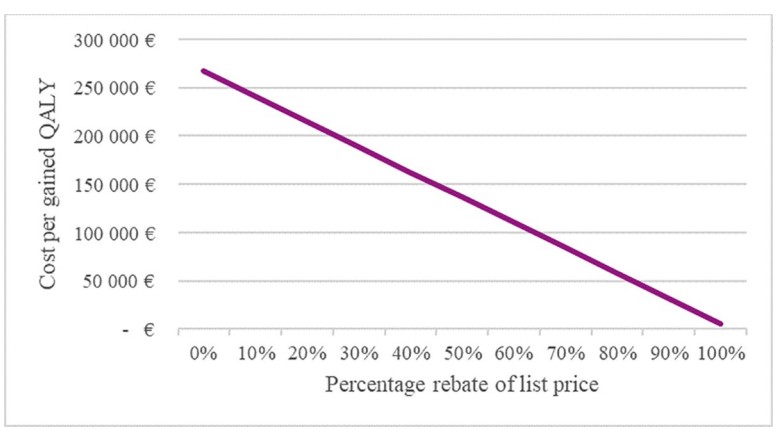

**Fig 6. Cost per gained QALY for the live attenuated herpes zoster vaccine for different prices of the vaccine.**

a societal perspective. The combined programme with both varicella and herpes zoster vaccination resulted in a cost per gained QALY of almost EUR 1.6 million, compared to only vaccinate against varicella, which cannot be considered cost-effective. There is no official threshold in Sweden for cost-effectiveness, but the lower the value of the ICER, the more favourable is the programme in terms of cost-effectiveness.

With varicella vaccination, there were only a few natural varicella cases by the end of the 85-year time horizon. Varicella vaccination also had a great impact on the herpes zoster incidence. At the end of the 85-year time horizon, the number of herpes zoster cases had decreased by 75%. The corresponding figure for the herpes zoster vaccine was a decrease with 2.3%. This means that the varicella vaccination would have a greater impact on the herpes zoster incidence than the herpes zoster vaccination, both on a 20-year time horizon as well as an 85-year time horizon. However, the number of herpes zoster cases will never go to zero; even with universal varicella vaccination and a 100% vaccination coverage, a few cases of herpes zoster will appear annually due to reactivation of the latent varicella vaccine virus [17, 55].

The assumed vaccination coverage was much lower for the herpes zoster vaccine than for the varicella vaccine, and the results from the sensitivity analyses suggest (see Table 5) that a higher vaccination coverage of herpes zoster vaccination would indicate a more favourable cost per gained QALY. Our assumptions about a moderate boosting efficacy of varicella exposure lead to a less pronounced temporary increase in herpes zoster incidence compared to many previous models, which also favours the cost-effectiveness of varicella vaccination. The uncertainty around the influence of varicella vaccination on herpes zoster incidence and reduced exogenous boosting has delayed decisions regarding the introduction of general varicella vaccination in several countries [56]. We have chosen a moderate approach to model exogenous boosting that made boosting comparable to the temporary protection of the live herpes zoster vaccination as described e.g. in Melegaro et al [18]. Thus, our results have a substantially lower surge in herpes zoster cases after varicella vaccination compared to previous VZV-modelling work. It should be stressed that the question of exogenous boosting cannot be resolved by modelling alone, but scientific progress on the immunological aspects and further observational and surveillance studies are needed before reasonable predictions of the impact of reduced boosting on herpes zoster incidence in the population can be made.

With, and without, the inclusion of production losses, varicella vaccination would be a dominating strategy, i.e. have a better effect at a lower cost. The inclusion of production losses had no significant impact on the results for the herpes zoster vaccination. What appears to have the greatest impact on the results for the vaccination programmes, were the discount rate and time horizon. In addidion, the sensitivity analyses that excluded the effect of exogenous boosting due to varicella vaccination on the herpes zoster incidence, had an impact on the results for the herpes zoster vaccination.

The results from the transmission model indicate a better effect of vaccinating with the recombinant vaccine, than with the live attenuated vaccine, against herpes zoster. The recombinant vaccine, Shingrix, was not yet marketed in Sweden as of March 2020, and we, therefore, had no information about the price in Sweden. However, a sensitivity analysis, assuming a price based on the USA market, which was close to the price of the recombinant vaccine on the Swedish private market, indicate that it would not be cost-effective in a Swedish setting, given the assumptions listed in this paper.

VZV-reactivations, e.g. meningoencephalitis, are not always linked to an outbreak of a herpes zoster rash and increasing evidence also suggests that varicella-zoster reactivations can be correlated to other conditions [57–59]. The extent of these non-rash reactivations was not yet established and were therefore not included in our incidence estimates.

The time horizon and discount rate had a great impact on the results, although not necessarily on the implications of the results. The fact that these were so influential can be problematic since changes in the society are likely to occur over time. This concerns both technological advances and changes in the health care sector at large. Modelled results can, therefore, be inaccurate over a longer period, which decision-makers need to be aware of. It is also important that decision-makers take the impact of the discount rate and that the effect of the childhood varicella vaccination programme on herpes zoster incidence is not immediate into account. As visible in Table 5, the results from a sensitivity analysis suggests that with a time horizon ranging from year 30 to year 50 after implementation of herpes zoster vaccination, the ICER decreased with almost 50%. This put forward that in a situation where we have reached a steady state after vaccine implementation, the cost per gained QALY would be much lower than in the implementation phase of the vaccination programme. The steady state results were in line with previous studies [60]. The results from the sensitivity analysis indicate a more favourable cost per gained QALY if vaccination against herpes zoster would target 75-year-olds, rather than 65-year-olds.

Due to the considerable differences between countries regarding their health care systems and price of vaccines, as well as discount rates and applied time horizon, it is problematic to compare results from cost-effectiveness studies between countries. However, our results from the health economic evaluation of varicella were in line with previous studies. A systematic review of the cost-effectiveness of routine varicella and herpes zoster vaccination for high-income countries from 2015 identified 23 model-based studies evaluating the cost-effectiveness of varicella vaccination [61]. Out of these studies, 13 used a dynamic modelling approach as our study did and accounted for herd immunity, and 10 used static models where one included an adjustment factor to account for herd immunity. The price of the vaccine varied widely between the studies, as did the length of the time horizon (20 years to a lifetime), and vaccination coverage. Varicella vaccination was found to be cost-effective or cost-saving (i.e. dominant) from a health care perspective and cost-saving from a societal perspective [61], in all the 23 studies when only varicella was considered, i.e. when herpes zoster was excluded, which was in line with the results from our study. However, when also herpes zoster was included and the effect of a reduced exogenous boosting due to varicella vaccination on the herpes zoster incidence, the cost-effectiveness was doubtful/unlikely. In our study, the effect of exogenous boosting was much lower than in previous studies, and the implication of the results from the base case did not change when exogenous boosting was excluded (Table 5).

The same systematic review as mentioned above identified 17 studies evaluating the cost-effectiveness of herpes zoster vaccination [61], where most of the studies were based on a static Markov-model using single or multiple cohorts. Most studies used a lifetime horizon, but the age at vaccination ranged between 50 and 80 years indicating a variation in time horizons. The cost of the vaccine differed widely between the studies (EUR 8–71), and the results from the studies ranged from EUR 1,200–291,240 per QALY in the health care perspective, and EUR 5,628–250,470 per QALY in the societal perspective [61]. Few of the studies in the review assumed a waning of the vaccine effectiveness close to our study–others assumed no waning or a very low waning rate, which could be a reason for the discrepancy of the results. Also, assumptions about vaccination coverage, age at vaccination, price of the vaccine, cost per case, time horizon and discount rate differ both among the studies in the review and compared to our study. Vaccination with the live herpes zoster vaccine was deemed cost-effective in several studies that were conducted before 2014 when the long-term follow-up study on the duration of vaccine effectiveness was published [61–63]. After 2014, studies instead apply a shorter duration, more in line with the duration applied in our study, which gave rise to less favourable cost-effectiveness estimates [64–66].

A systematic review of cost-effectiveness studies of herpes zoster vaccination from 2019 [63] state that 15 out of 25 studies found that the live attenuated vaccine was cost-effective in comparison to no vaccination, with a vaccine price that ranged between US$93 and US$236 per dose. In the like with the systematic review from 2015 [61], the ingoing studies varied greatly both among the studies and in comparison to our study, with regards to vaccination coverage, age at vaccination, waning rate of the vaccine, price of the vaccine, cost per case, time horizon, and discount rate and the included health states. A systematic review from 2014 of the cost-effectiveness of herpes zoster vaccination [62] concludes that the majority of the included 14 cost-effectiveness studies found herpes zoster vaccination to be cost-effective compared to no vaccination. The ICER varied between EUR 2,300 (a best-case scenario) to EUR 104,000. However, many of the deductions of cost-effectiveness were contingent on a specific price of the vaccine or a vaccine duration that last longer than 10 years. In general, the studies assuming a life-long duration gave rise to the lowest ICER. This was in contrast to our study, where we do not conclude that cost-effectiveness of herpes zoster vaccine would be cost-effective given a longer duration of the vaccine, based on now available data on duration of the live vaccine. The impact of the vaccine price on the ICER illustrated in Fig 5. The cost-per gained QALY of the studies in the review varied greatly depending on the target population, model structure, and the ingoing parameters. The review also concluded that factors that were influential to the ICER were duration of vaccine efficacy, vaccine price, age at vaccination, and the applied discount rate.

Previous studies have found that vaccination against herpes zoster with the recombinant vaccine was likely to be cost-effective [67], and the results were reliant on the vaccine having a great impact on the burden of illness from herpes zoster. In our population based-model, and given the assumption stated in the previous parts of this paper, the impact did not appear as significant as in previous studies.

Our primary objective was to investigate the cost-effectiveness of vaccination programmes targeting the varicella-zoster virus in Sweden. Compared to no vaccination, the varicella vaccination programme was the most cost-effective of the three investigated scenarios. We can conclude that out of a health economic perspective the most reasonable outcome would be an introduction of varicella vaccination with or without herpes zoster vaccination in a Swedish national vaccination programme. The findings are important since Sweden was currently assessing varicella and herpes zoster vaccination policies and whether the diseases should be included in a national vaccination programme. Our results will help guide decision-makers to make informed decisions on the most effective way to implement such a programme.

The waning rate, or rather the duration of vaccine protection, after two doses of varicella vaccine, was uncertain in the long-term [68] and long-term impact and immunity needs to be followed. Should an unexpectedly large rise in herpes zoster occur 10 to 20 years after initiation of childhood vaccination, herpes zoster vaccination could be re-evaluated for cost-effectiveness. There will always be a few cases of herpes zoster in Sweden from reactivation of latent varicella vaccine virus in vaccinated individuals [17, 55]. Also, the link between herpes zoster and sequelae needs to be studied to include these suitably in health economic evaluations. As of now, it was hard to include all of them since the attributable fraction to herpes zoster and other reactivations of virus was unclear.

## Conclusion

The results from the health economic modelling suggest that it was cost-effective to introduce varicella vaccination in Sweden, whereas herpes zoster vaccination with the live vaccine for the elderly was not cost-effective–the health effects of the latter vaccination cannot be considered

reasonable in relation to its costs. The results from this study are central components for policy-relevant decision-making.

## Author Contributions

**Conceptualization:** Ellen Wolff, Katarina Widgren, Gianpaolo Scalia Tomba.

**Data curation:** Ellen Wolff, Katarina Widgren, Gianpaolo Scalia Tomba, Tiia Lep, Sören Andersson.

**Formal analysis:** Ellen Wolff, Katarina Widgren.

**Investigation:** Ellen Wolff.

**Methodology:** Ellen Wolff, Katarina Widgren, Gianpaolo Scalia Tomba.

**Supervision:** Adam Roth.

**Validation:** Ellen Wolff.

**Writing – original draft:** Ellen Wolff.

**Writing – review & editing:** Katarina Widgren, Gianpaolo Scalia Tomba, Adam Roth, Tiia Lep, Sören Andersson.

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
