## [Decision Letter · Decision Letter 0]

7 Oct 2020

PONE-D-20-26903

Cost-effectiveness of varicella and herpes zoster vaccination in Sweden: an economic evaluation using a dynamic transmission model

PLOS ONE

Dear Dr. Wolff,

Thank you for submitting your manuscript to PLOS ONE. After careful consideration, we feel that it has merit but does not fully meet PLOS ONE’s publication criteria as it currently stands. Therefore, we invite you to submit a revised version of the manuscript that addresses the points raised during the review process.

Please submit your revised manuscript within 3 months from now. If you will need more time than this to complete your revisions, please reply to this message or contact the journal office at plosone@plos.org. Please include the following items when submitting your revised manuscript:

We look forward to receiving your revised manuscript.

Kind regards,

Georges M.G.M. Verjans, MSc, PhD

Academic Editor

PLOS ONE

Additional Editor Comments:

The MS submitted by Wolff E. et al. describes the use of a mathematical model to determine if varicella and/of herpes zoster vaccination is cost-effective to be introduced in the national vaccination program of Sweden. The study is well designed and of interest, but needs significant improvement for the general reader to appreciate the modelling applied and to discuss the data obtained in more detail.

Journal Requirements:

2. Please amend your manuscript to include your abstract after the title page.

Reviewers' comments:

Reviewer's Responses to Questions

**Comments to the Author**

1. Is the manuscript technically sound, and do the data support the conclusions?

Reviewer #1: Yes

Reviewer #2: Partly

2. Has the statistical analysis been performed appropriately and rigorously? 

Reviewer #1: Yes

Reviewer #2: Yes

3. Have the authors made all data underlying the findings in their manuscript fully available?

Reviewer #1: Yes

Reviewer #2: No

4. Is the manuscript presented in an intelligible fashion and written in standard English?

Reviewer #1: Yes

Reviewer #2: No

5. Review Comments to the Author

Reviewer #1: The authors should clearly state that zoster can be caused by the live attenuated varicella vaccine virus. This point is never clearly stated anywhere in the manuscript. Suggest 2 additional references be added at these locations. [1] Discussion, line 301. "The number of cases of herpes zoster will never go to zero, even after universal varicella vaccination. There will always be a few cases of herpes zoster caused by reactivation of latent varicella vaccine virus. " Please cite 2 references: PMID: 30496366 by Harpaz (2019) and PMID: 32677551 by Ramachandran (2020). [2] Discussion, lines 417-8. "There will always be a low background of herpes zoster in Sweden from reactivation of latent varicella vaccine virus in immunized citizens." Cite same 2 references.

Reviewer #2: In the report by Wolff et al submitted to PLOS One a predictive study is presented from a number of assumptions that models the cost effectiveness of varicella and HZ vaccination in Sweden, using a rather not well explained or fully documented dynamic transition model. Based on the assumptions and modeling the authors conclude that the varicella vaccination program has a much stronger cost effectiveness, when given with or without the live zoster vaccination program. The live zoster vaccination program seems to only have a mild effect on incidence and cost effectiveness that leads them to conclude it is not cost effective. This is different from most of the cost effectiveness studies from other countries, which generally indicate a zoster vaccine would be cost effective.

This review is given by a person knowledgeable in the field of varicella and zoster who is not a statistical epidemiologist familiar with most of the details. As such, the issues I raise may be considered representative of those that might be raised by other readers.

1. In the introduction, please carefully explain what Sweden’s current vaccination policies are with both varicella and zoster vaccines. Are one or two dose regimens mandated for varicella? what is the coverage rate? , What is the status of the live zoster vaccine and its coverage, specifically in Sweden? It is indicated for use in 65-year-old, but this is very different from the US where it was 60, then 50. Indeed, does dosing at 50 years affect the analyses? Does increasing coverage (what is it currently?) result in different conclusions? Is it mandated? None of this would be familiar to the reader from the manuscript as is currently written.

2. A key reference to many of the assumptions is reference 10. This cannot be looked up and is incomplete and I quote “10. Folkhälsomyndigheten. VZV vaccine - EDIT! ; 2019.”!!!!!! This model could do with a more through description than given and should be a stand-alone description that does not require this reference. Is it being, lines 62 and following regarding the model don’t always make sense to the non-expert reader?

3. Grammar corrections are needed throughout the manuscript. Just as one of many examples, lines 40-41 have 7 commas. Effective vaccines against varicella (line 39 have not been “available” since the mid-1980s but rather since the mid-1990s.

4. Line 86, varicella vaccinees are also at risk for subclinical infection and reactivation with WT zoster. This may be low. Line 99, what is meant by choosing a moderate approach to include exogenous boosting, comparable to HZ live vaccination, when the documentation for this is rather vague? This is a rather large leap of faith that may not be correct.

5. Given that Finland’s data is also used for the analyses, the programs of vaccination in that country are also important to document, particularly if they differ from that of Sweden

6. A number of the assumptions could do with strengthening from US data. For example, 116, the risk of vaccine virus being 10% seems to be rather different from the accepted US HZ data on zoster in vaccinees, which rather implies that vaccine virus is considerably rarer than 10%. Likewise, the PHN and non PHN assumptions could be substantiated better from US data. The coverage of vaccination at 95% seems to be rather high and more than seem in most countries with a mandated varicella vaccination program I do not believe the US rates are not this high, and I am certain that accurate coverage data should be available and used rather than assumptions?

7. Line 156-7 if varicella breakthrough occurs in a VZV vaccinee there is (logically) a higher rate of physician’s visits, incurring costs, no?

8. Line 174 it is not clear why an extra physician’s visit was required for the administration of the vaccine., please explain

9. As a non-expert reviewer, please explain better the concepts of sensitivity and discount rates line 212

10. The zoster vaccine is given at 65, what is the influence of giving the vaccine at age 50 as occurred in the US? Also, it is not obvious why the model does not show a reduction in zoster rates in Figure 2 in zoster vaccinees. I really tried to follow their arguments for this and it was not at all clear. It is well established from the trials that HZ incidence is reduced by HZ vaccination, why is that not seen here? Is this largely due to low vaccine coverage and partial protection rates? I would have expected a greater influence, particularly towards the end of the 100-year age spectrum.

6. PLOS authors have the option to publish the peer review history of their article (what does this mean?). If published, this will include your full peer review and any attached files.

Reviewer #1: No

Reviewer #2: No

---

## [Author Response · Author response to Decision Letter 0]

29 Mar 2021

Detailed response to reviwers comments

Ellen Wolff

November 2020

Firstly, we would like to thank the reviewers for the very good and insightful comments that add a lot to the paper.

Reviewer 1: 

Comments to reviewer 1.

1. The authors should clearly state that zoster can be caused by the live attenuated varicella vaccine virus. This point is never clearly stated anywhere in the manuscript. Suggest 2 additional references be added at these locations. Discussion, line 301. "The number of cases of herpes zoster will never go to zero, even after universal varicella vaccination. There will always be a few cases of herpes zoster caused by reactivation of latent varicella vaccine virus. " Please cite 2 references: 1by Harpaz (2019) and by Ramachandran (2020). Discussion, lines 417-8. "There will always be a low background of herpes zoster in Sweden from reactivation of latent varicella vaccine virus in immunized citizens." Cite same 2 references.

Reply: Thank you for this comment. It has now been added together with the references to the manuscript. See the discussion-section, paragraph 2, and the last paragraph. 

Reviewer 2:

In the report by Wolff et al submitted to PLOS One a predictive study is presented from a number of assumptions that models the cost effectiveness of varicella and HZ vaccination in Sweden, using a rather not well explained or fully documented dynamic transition model. Based on the assumptions and modeling the authors conclude that the varicella vaccination program has a much stronger cost effectiveness, when given with or without the live zoster vaccination program. The live zoster vaccination program seems to only have a mild effect on incidence and cost effectiveness that leads them to conclude it is not cost effective. This is different from most of the cost effectiveness studies from other countries, which generally indicate a zoster vaccine would be cost effective.

This review is given by a person knowledgeable in the field of varicella and zoster who is not a statistical epidemiologist familiar with most of the details. As such, the issues I raise may be considered representative of those that might be raised by other readers.

Comment to Reviewer 2

1. In the introduction, please carefully explain what Sweden’s current vaccination policies are with both varicella and zoster vaccines. Are one or two dose regimens mandated for varicella? what is the coverage rate? , What is the status of the live zoster vaccine and its coverage, specifically in Sweden? It is indicated for use in 65-year-old, but this is very different from the US where it was 60, then 50. Indeed, does dosing at 50 years affect the analyses? Does increasing coverage (what is it currently?) result in different conclusions? Is it mandated? None of this would be familiar to the reader from the manuscript as is currently written

Reply: Thank you for this comment, it is now clearly stated in the introduction that neither varicella not herpes zoster is part of a national vaccination programme in Sweden but that doses are available on the private market, see the introduction section, the last paragraph.

2. A key reference to many of the assumptions is reference 10. This cannot be looked up and is incomplete and I quote “10. Folkhälsomyndigheten. VZV vaccine - EDIT! ; 2019.”!!!!!! This model could do with a more through description than given and should be a stand-alone description that does not require this reference. Is it being, lines 62 and following regarding the model don’t always make sense to the non-expert reader?

Reply: Unfortunately, the paper we would have preferred to cite in not yet published. Therefore, we have added a more thorough description of the epidemiological model. See ”The Epidemiological Model”-section.

3. Grammar corrections are needed throughout the manuscript. Just as one of many examples, lines 40-41 have 7 commas. Effective vaccines against varicella (line 39 have not been “available” since the mid-1980s but rather since the mid-1990s.

Reply: We have revised the manuscript and corrected grammar mistakes. Changes have been made throughout the manuscript. We also changed the sentence to “Effective vaccines against varicella have been in use since the mid-1990s...” Thank you for noticing this. 

4. Line 86, varicella vaccinees are also at risk for subclinical infection and reactivation with WT zoster. This may be low. Line 99, what is meant by choosing a moderate approach to include exogenous boosting, comparable to HZ live vaccination, when the documentation for this is rather vague? This is a rather large leap of faith that may not be correct.

Reply: Thank you, this has been added to the manuscript, see section “The Epidemilogical Model”, the last paragraph, and was also accounted for in the model. 

5. Given that Finland’s data is also used for the analyses, the programs of vaccination in that country are also important to document, particularly if they differ from that of Sweden

Reply: This is a very important point and information has now been added to the manuscript, see “Incidence of varicella”-section. 

6. A number of the assumptions could do with strengthening from US data. For example, 116, the risk of vaccine virus being 10% seems to be rather different from the accepted US HZ data on zoster in vaccinees, which rather implies that vaccine virus is considerably rarer than 10%. Likewise, the PHN and non PHN assumptions could be substantiated better from US data. The coverage of vaccination at 95% seems to be rather high and more than seem in most countries with a mandated varicella vaccination program I do not believe the US rates are not this high, and I am certain that accurate coverage data should be available and used rather than assumptions?

Reply: We have added references to the manuscript to strengthen our assumptions.

Varicella vaccination would be part of the national vaccination programme for children in Sweden. The programme has a coverage rate of about 98% among 2-year olds even though it is not a mandatory programme. Therefore, we believe that reaching a 95% coverage rate is reasonable. We have also included a sensitivity analysis with a coverage of 85%.

7. Line 156-7 if varicella breakthrough occurs in a VZV vaccinee there is (logically) a higher rate of physician’s visits, incurring costs, no?

Reply: For patients with breakthrough varicella we did not assume that they would be hospitalized, but physician’s visits are included in the analyses. No changes have been made to the model, but the cost-section has been updated and clarified. 

8. Line 174 it is not clear why an extra physician’s visit was required for the administration of the vaccine., please explain

Reply: This has been clarified in the manuscript; see the resource use-section, the fourth paragraph. As of today, there is no vaccination programme in Sweden targeting people over 65 years old, for instance for seasonal influenza. Therefore, there is no natural reason to visit a physician and get the herpes zoster vaccine without needing an extra visit for the vast majority.

9. As a non-expert reviewer, please explain better the concepts of sensitivity and discount rates line 212.

Reply: This has been clarified in the manuscript, see the sensitivity analyses-section, the first paragraph.

10. The zoster vaccine is given at 65, what is the influence of giving the vaccine at age 50 as occurred in the US? Also, it is not obvious why the model does not show a reduction in zoster rates in Figure 2 in zoster vaccinees. I really tried to follow their arguments for this and it was not at all clear. It is well established from the trials that HZ incidence is reduced by HZ vaccination, why is that not seen here? Is this largely due to low vaccine coverage and partial protection rates? I would have expected a greater influence, particularly towards the end of the 100-year age spectrum.

Reply: We chose to model the ages where vaccination potentially has the greatest effect. Due to the incidence increasing with age and the short duration of the vaccine. We modelled vaccination at 65 years and run a sensitivity analysis for vaccination at 75 years. We found that vaccination at 75 years was the most cost-effecktive, biut neither this programme was dominant. Also, it is not realistic that a vaccination programme against herpes zoster would target 50-year-olds in Sweden and the expected effect would be lower. 

Figure 3 that shows the herpes zoster incidence with different vaccination strategies, has been updated to more clearly show the effects of vaccination. The small decrease in incidence is partly due to the short duration of the vaccine, vaccination coverage, and effect of the vaccine. The main reason for the modest impact is that we do not show the impact on the vaccinated cohort, but rather the impact on the zoster incidence in the entire population.

---

## [Editor Report · Decision Letter 1]

7 Apr 2021

PONE-D-20-26903R1

Cost-effectiveness of varicella and herpes zoster vaccination in Sweden: an economic evaluation using a dynamic transmission model

PLOS ONE

Dear Dr. Wolff,

Thank you for submitting your manuscript to PLOS ONE. After careful consideration, we feel that it has merit but does not fully meet PLOS ONE’s publication criteria as it currently stands. Therefore, we invite you to submit a revised version of the manuscript that addresses the points raised during the review process.

We look forward to receiving your revised manuscript.

Kind regards,

Georges M.G.M. Verjans, MSc, PhD

Academic Editor

PLOS ONE

Additional Editor Comments (if provided):

The authors submitted a revised manuscript (one clean and one with track changes) and their response to reviewers' comments.

Before considering this revised manuscript, this Editor appreciates the authors to take care of two critical issues:

1. Manuscripts with track changes are very difficult to interpret. It is appreciated to receive a 'marked copy of the manuscript' in which the changes are indicated in 'red'. This needs to be accompanied with an updated authors' response to reviewers' comments with detailed info which text is changed, including reference to 'line numbers'.

2. Furthermore, changes to authorship have been made. This is a MAJOR ISSUE that needs to be clarified and approved by the editorial office. To request such change, this Editor appreciates to receive the following from the corresponding author: (a) the reason for the change in author list and (b) written confirmation (e-mail, letter) from all authors that they agree with the author rearrangement.

---

## [Author Response · Author response to Decision Letter 1]

28 Apr 2021

aA detailed response to reviewers comments

Ellen Wolff

November 2020

Firstly, we would like to thank the reviewers for the very good and insightful comments that add a lot to the paper. References to line numbers consider the version of the manuscript named “marked copy”,

Reviewer 1: 

Comments to reviewer 1.

1. The authors should clearly state that zoster can be caused by the live attenuated varicella vaccine virus. This point is never clearly stated anywhere in the manuscript. Suggest 2 additional references be added at these locations. Discussion, line 301. "The number of cases of herpes zoster will never go to zero, even after universal varicella vaccination. There will always be a few cases of herpes zoster caused by reactivation of latent varicella vaccine virus. " Please cite 2 references: 1by Harpaz (2019) and by Ramachandran (2020). Discussion, lines 417-8. "There will always be a low background of herpes zoster in Sweden from reactivation of latent varicella vaccine virus in immunized citizens." Cite same 2 references.

Reply: Thank you for this comment. It has now been added together with the references to the manuscript. See the discussion-section, lines 383-386, and 499-501

Reviewer 2:

In the report by Wolff et al submitted to PLOS One a predictive study is presented from a number of assumptions that models the cost effectiveness of varicella and HZ vaccination in Sweden, using a rather not well explained or fully documented dynamic transition model. Based on the assumptions and modeling the authors conclude that the varicella vaccination program has a much stronger cost effectiveness, when given with or without the live zoster vaccination program. The live zoster vaccination program seems to only have a mild effect on incidence and cost effectiveness that leads them to conclude it is not cost effective. This is different from most of the cost effectiveness studies from other countries, which generally indicate a zoster vaccine would be cost effective.

This review is given by a person knowledgeable in the field of varicella and zoster who is not a statistical epidemiologist familiar with most of the details. As such, the issues I raise may be considered representative of those that might be raised by other readers.

Comment to Reviewer 2

1. In the introduction, please carefully explain what Sweden’s current vaccination policies are with both varicella and zoster vaccines. Are one or two dose regimens mandated for varicella? what is the coverage rate? , What is the status of the live zoster vaccine and its coverage, specifically in Sweden? It is indicated for use in 65-year-old, but this is very different from the US where it was 60, then 50. Indeed, does dosing at 50 years affect the analyses? Does increasing coverage (what is it currently?) result in different conclusions? Is it mandated? None of this would be familiar to the reader from the manuscript as is currently written

Reply: Thank you for this comment, it is now clearly stated in the introduction that neither varicella not herpes zoster is part of a national vaccination programme in Sweden but that doses are available on the private market, see the introduction section, lines 87-89.

2. A key reference to many of the assumptions is reference 10. This cannot be looked up and is incomplete and I quote “10. Folkhälsomyndigheten. VZV vaccine - EDIT! ; 2019.”!!!!!! This model could do with a more through description than given and should be a stand-alone description that does not require this reference. Is it being, lines 62 and following regarding the model don’t always make sense to the non-expert reader?

Reply: Unfortunately, the paper we would have preferred to cite is not yet published. Therefore, we have added a more thorough description of the epidemiological model. See ”The Epidemiological Model”-section, line 106-110, 114-118, and 140-141.

3. Grammar corrections are needed throughout the manuscript. Just as one of many examples, lines 40-41 have 7 commas. Effective vaccines against varicella (line 39 have not been “available” since the mid-1980s but rather since the mid-1990s.

Reply: We have revised the manuscript and corrected grammar mistakes. Changes have been made throughout the manuscript. We also changed the sentence to “Effective vaccines against varicella have been in use since the mid-1990s...” Thank you for noticing this. 

4. Line 86, varicella vaccinees are also at risk for subclinical infection and reactivation with WT zoster. This may be low. Line 99, what is meant by choosing a moderate approach to include exogenous boosting, comparable to HZ live vaccination, when the documentation for this is rather vague? This is a rather large leap of faith that may not be correct.

Reply: Thank you, this has been added to the manuscript, see section “The Epidemiological Model”, the last paragraph, lines 140-141, 144-146. 

5. Given that Finland’s data is also used for the analyses, the programs of vaccination in that country are also important to document, particularly if they differ from that of Sweden

Reply: This is a very important point and information has now been added to the manuscript, see “Incidence of varicella”-section, lines 150-151.

6. A number of the assumptions could do with strengthening from US data. For example, 116, the risk of vaccine virus being 10% seems to be rather different from the accepted US HZ data on zoster in vaccinees, which rather implies that vaccine virus is considerably rarer than 10%. Likewise, the PHN and non PHN assumptions could be substantiated better from US data. The coverage of vaccination at 95% seems to be rather high and more than seem in most countries with a mandated varicella vaccination program I do not believe the US rates are not this high, and I am certain that accurate coverage data should be available and used rather than assumptions?

Reply: We have added references to the manuscript to strengthen our assumptions.

Varicella vaccination would be part of the national vaccination programme for children in Sweden. The programme has a coverage rate of about 98% among 2-year olds even though it is not a mandatory programme. Therefore, we believe that reaching a 95% coverage rate is reasonable. We have also included a sensitivity analysis with a coverage of 85%.

7. Line 156-7 if varicella breakthrough occurs in a VZV vaccinee there is (logically) a higher rate of physician’s visits, incurring costs, no?

Reply: For patients with breakthrough varicella we did not assume that they would be hospitalized, but physician’s visits are included in the analyses. No changes have been made to the model, but the cost section has been updated and clarified. 

8. Line 174 it is not clear why an extra physician’s visit was required for the administration of the vaccine., please explain

Reply: This has been clarified in the manuscript; see the resource use-section, line 218-220. As of today, there is no vaccination programme in Sweden targeting people over 65 years old, for instance for seasonal influenza. Therefore, there is no natural reason to visit a physician and get the herpes zoster vaccine without needing an extra visit for the vast majority.

9. As a non-expert reviewer, please explain better the concepts of sensitivity and discount rates line 212.

Reply: This has been clarified in the manuscript, see the sensitivity analyses section, lines 277-280.

10. The zoster vaccine is given at 65, what is the influence of giving the vaccine at age 50 as occurred in the US? Also, it is not obvious why the model does not show a reduction in zoster rates in Figure 2 in zoster vaccinees. I really tried to follow their arguments for this and it was not at all clear. It is well established from the trials that HZ incidence is reduced by HZ vaccination, why is that not seen here? Is this largely due to low vaccine coverage and partial protection rates? I would have expected a greater influence, particularly towards the end of the 100-year age spectrum.

Reply: We chose to model the ages where vaccination potentially has the greatest effect. Due to the incidence increasing with age and the short duration of the vaccine. We modeled vaccination at 65 years and run a sensitivity analysis for vaccination at 75 years. We found that vaccination at 75 years was the most cost-effective, but the programme was dominant. Also, it is not realistic that a vaccination programme against herpes zoster would target 50-year-olds in Sweden and the expected effect would be lower. 

Figure 3, which shows the herpes zoster incidence with different vaccination strategies, has been updated to more clearly show the effects of vaccination. The small decrease in incidence is partly due to the short duration of the vaccine, vaccination coverage, and effect of the vaccine. The main reason for the modest impact is that we do not show the impact on the vaccinated cohort, but rather the impact on the zoster incidence in the entire population.

---

## [Editor Report · Decision Letter 2]

30 Apr 2021

Cost-effectiveness of varicella and herpes zoster vaccination in Sweden: an economic evaluation using a dynamic transmission model

PONE-D-20-26903R2

Dear Dr. Wolff,

We’re pleased to inform you that your manuscript has been judged scientifically suitable for publication and will be formally accepted for publication once it meets all outstanding technical requirements.

Kind regards,

Georges M.G.M. Verjans, MSc, PhD

Academic Editor

PLOS ONE

Additional Editor Comments (optional):

Authors revised the manuscript satisfactory by addressing the reviewers' critiques sufficiently.

Reviewers' comments:

Not applicable.

---

## [Editor Report · Acceptance letter]

4 May 2021

PONE-D-20-26903R2 

Cost-effectiveness of varicella and herpes zoster vaccination in Sweden: an economic evaluation using a dynamic transmission model 

Dear Dr. Wolff:

I'm pleased to inform you that your manuscript has been deemed suitable for publication in PLOS ONE. Congratulations! Your manuscript is now with our production department. 

Kind regards, 

on behalf of

Prof. Dr. Georges M.G.M. Verjans 

Academic Editor

PLOS ONE